# CircPRMT5 promotes progression of osteosarcoma by recruiting CNBP to regulate the translation and stability of CDK6 mRNA

Yunlu Liu[1◉], Hongyan Jiang[2◉], Keli Hu[1], Hui Zou[1], Weiguo Zhang[1], Jiangtao Liu[1], Xiaofei Jian[1]*

1 Department of Orthopedics, The Central Hospital of Wuhan, Tongji Medical College, Huazhong University of Science and Technology, Wuhan, 430014, China, 2 Department of Otorhinolaryngology, Union Hospital, Tongji Medical College, Huazhong University of Science and Technology, Wuhan, 430022, China

◉ These authors contributed equally to this work.
* xiaofeijian0417@126.com

**Data Availability Statement:** All relevant data are within the manuscript and its Supporting Information files.

## Abstract

Research has demonstrated that circular RNAs (circRNAs) exert critical functions in the occurrence and progression of numerous malignant tumors. CircPRMT5 was recently reported to be involved in the pathogenesis of cancers. However, the potential role of circPRMT5 in osteosarcoma needs further investigation. In present study, our results suggested that circPRMT5 was highly upregulated in osteosarcoma cells and mainly localizes in the cytoplasm. CircPRMT5 promoted the proliferation, migration and invasion capacities of osteosarcoma cells, and suppressed cell apoptosis. Knockdown of circPRMT5 exerted the opposite effects. Mechanically, circPRMT5 promoted the binding of CNBP to CDK6 mRNA, which enhanced the stability of CDK6 mRNA and facilitated its translation, thereby promoting the progression of osteosarcoma. Knockdown of CDK6 reversed the promoting effect of circPRMT5 on osteosarcoma cells. These findings suggest that circPRMT5 promotes osteosarcoma cell malignant activity by recruiting CNBP to regulate the translation and stability of CDK6 mRNA. Thus, circPRMT5 may represent a promising therapeutic target for osteosarcoma.

## Introduction

As the most prevalent malignant bone neoplasm, osteosarcoma is a great threat to human health, especially for adolescents and children [1]. Osteosarcoma has a rapid progression, and many patients are already in the advanced stage when they are initially diagnosed [1,2]. The 5-year survival rate for osteosarcoma patients with recurrent or metastatic tumors is only 20% [3]. Currently, there is no effective treatment to stop osteosarcoma progression. Therefore, a deeper understanding of the molecular mechanisms underlying the pathogenesis of osteosarcoma is essential to identify new therapeutic targets.

Circular RNAs (circRNAs), formed by the reverse splicing of precursor mRNA exons [4,5], play a vital role in various human diseases, including renal diseases [6], rheumatoid arthritis

**Funding:** This work was supported by the Hubei Provincial Natural Science Foundation of China (Grant No. 2023AFB1014).

**Competing interests:** The authors have declared that no competing interests exist.

[7] and immune diseases [8]. Studies have also demonstrated a functional role of circRNAs in the pathophysiology of cancer [9]. For example, Cao et al. found that circRNF20 facilitated tumorigenesis of breast cancer via sponging miR-487a, thereby regulating the expression of hexokinase II [10]. Yang et al. reported that hsa_circRNA_0088036 promotes bladder cancer progression by regulating FOXQ1 expression [11]. Moreover, circRNAs have been considered potential targets for cancer diagnosis or therapy because of their structural stability, evolutionary conserved nature and organ specificity [5].

Previous researches have reported that circRNAs exert key biological functions in cancer development and progression by serving as miRNA sponges [12–14] or translation templates [15,16]. Emerging research has revealed that circRNAs can also bind RNA binding proteins (RBPs) to regulate the stability or translation of mRNAs [17]. For example, hsa_circ_0068631 was found to participate into the development of breast cancer by recruiting EIF4A3 to stabilize c-Myc mRNA [18]. Sun et al. found that circMYBL2 recruits PTBP1 to enhance the translation of FLT3, which plays a critical function in acute myeloid leukemia progression [19].

CircPRMT5 has been found to be upregulated in different types of cancers, including bladder urothelial cancer [20], thyroid cancer [21] and colorectal cancer [22], and exert vital functional roles in their progression. However, the role of circPRMT5 in osteosarcoma is not yet known. Here, we found that circPRMT5 expression levels in osteosarcoma cells was significantly upregulated, and overexpression of circPRMT5 significantly promoted the proliferation, invasion and metastasis of osteosarcoma cells. Mechanistically, circPRMT5 enhanced cyclin-dependent kinase 6 (CDK6) expression by promoting the binding of CCHC-type zinc finger nucleic acid binding protein (CNBP) to CDK6 mRNA in osteosarcoma cells. Together, these findings indicate that circPRMT5 may serve as an oncogenic factor to promote CNBP-facilitated CDK6 expression and osteosarcoma progression.

## Materials and methods

### Cell culture

The osteosarcoma cell lines and normal human osteoblasts (hFOB1.19) were purchased from Procell Life Science and Technology Co., Ltd. (Wuhan, China). The cells were cultured in appropriate medium containing 10% fetal bovine serum (FBS; Gibco, Grand Island, USA) at 37˚C in an incubator with 5% $CO_2$.

### Quantitative RT-PCR (qRT-PCR)

Total RNAs of the cultured cells were isolated with Trizol reagent. A nuclear and cytoplasm RNA extraction kit (RiboBio, Guangzhou) was used to analyze RNAs in nuclear and cytoplasmic extracts. qRT-PCR was performed using the SYBR Green PCR kit (TaKaRa, Kyoto, Japan) as previously reported [23]. The primers are shown in S1 Table.

### Cell transfection

Oligonucleotides of circPRMT5, sh-circPRMT5, sh-CDK6 and their negative controls were synthesized by GeneChem (Shanghai, China). These plasmids were transfected into cells for gene overexpression and knockdown using Lipofectamine 3000 (Invitrogen, MA, USA). The sequences are shown in S2 Table.

### RNA-fluorescence in situ hybridization (RNA-FISH)

Assays were performed using RNA-FISH kit (RiboBio, China) following the manufacturer's instructions. Biotin-labeled probe for circPRMT5 was synthesized by RiboBio Technology

Co., Ltd. The sequence is as follows: CircPRMT5: 5′-ACCCGCATCCAGAACTTGAGGAG CCGG-3′. Images were acquired using laser confocal microscopy (Leica, Germany).

## Cell proliferation assay

Cell viability was detected using the MTT kit (Proteintech, Wuhan, China). Briefly, 2,000 cells/ well were seeded into 96-well plates and cultured for various time periods. 10 μL of MTT reagent was added into the plates and incubated for 2h. Finally, the optical densities were measured at 570 nm.

## Colony formation assay

The cells were plated $1 \times 10^3$ cells/well and cultured under standard conditions for 2 weeks. After washing 2 times with PBS, the cell colonies were fixed by 10% formaldehyde and stained with 0.1% crystal violet.

## Transwell assay

Transwell assay was conducted using a Transwell chamber (Corning, Shanghai, China) as described previously [24]. Briefly, the indicated numbers of serum-free cell suspensions were plated into the top chamber. While, the complete medium with 20% FBS was added to the bottom chamber. After incubation, the cells were fixed with 10% formaldehyde and stained with 0.1% crystal violet.

## Wound-healing assay

The migration ability of osteosarcoma cells was evaluated using a wound-healing assay as described previously [25]. A density of $1.5 \times 10^5$ cells was added into a 6-well plate, and cultured for 24 h. Subsequently, a sterile plastic micropipette tips were used to create a straight wound of equal width at the bottom of the well. After 0 h and 48 h, images were taken with a light microscope.

## Western blot (WB) analysis

WB analysis was conducted as described previously [24]. Briefly, the protein samples were separated by 10% SDS-PAGE electrophoresis and transferred onto PVDF membranes (Millipore, MA, USA). The protein bands were probed with indicated antibodies and detected using an enhanced chemiluminescence system (Millipore). Antibody information is listed in S3 Table.

## TUNEL assay

The apoptosis of osteosarcoma cells was detected using the TUNEL cell apoptosis assay kit (Thermo Fisher Scientific, USA) according to the manufacturer's manual.

## Cell cycle analysis

The PI/RNase Staining Solution kit (Thermo) was used to assess the cell cycle distribution. Briefly, the cells were collected and fixed with cold 70% ethanol. After staining with 50 μg/ml PI and 20 μg/ml RNase solution, subsequently the cells were subjected to flow cytometry.

### RNA-immunoprecipitation (RIP)

The RIP assays were performed with the RIP kit (Millipore). Briefly, the cells were lysed and incubated with target antibody overnight or IgG. The co-precipitated RNAs was extracted and analyzed by qRT-PCR. The antibody information for RIP is presented in S3 Table.

### RNA pull-down

Assays were preformed using the RNA-protein pull-down kit (Thermo). The protein complex immunoprecipitated by circPRMT5 was eluted and subjected to WB analysis. The gel was stained by Coomassie bright blue (Merck, Germany). Mass spectrometry was used to identify the proteins immunoprecipitated by circPRMT5.

### Bioinformatic analysis

The mass spectrum (MS) results for circPRMT5-interacting factors were overlapped with RBP and transcription factor (TF) databases. The Cancer Genome Atlas (TCGA) database and Gene Expression Omnibus (GEO) database (accession number: GSE12865) were used to identify the differentially expressed mRNAs between normal and osteosarcoma tissues. The Encyclopedia of RNA Interactomes (ENCORI) database was used to predict the target mRNAs of circPRMT5.

### Statistical analysis

Statistical analyses were performed using GraphPad Prism 8.0 software. All data are expressed as mean ± standard deviation (SD) with the difference compared by student's $t$-test or one-way analysis of variance. P values $< 0.05$ were considered significant.

## Results

### CircPRMT5 is highly expressed in osteosarcoma cells

CircPRMT5 has been implicated in tumor initiation and progression [20,22]. However, its role in osteosarcoma has not been reported. First, we performed Sanger sequencing on the PCR product of circPRMT5 in osteosarcoma cells and verified the cyclization site (Fig 1A). Analysis of amplification results with convergent or divergent primers confirmed the circular structure of circPRMT5 (Fig 1B). We then examined the circPRMT5 expression in different osteosarcoma cell lines, and found that circPRMT5 was upregulated in MNNG/HOS and MG-63 cells (Fig 1C). Thus, we selected these two cell lines for subsequent experiments. The results of RNase R exonuclease digestion experiments showed that circPRMT5 was resistant to RNase R exonuclease (Fig 1D and 1E). Additionally, circPRMT5 had a longer half-life than PRMT5 (Fig 1F and 1G). These results indicated the stable circular structure of circPRMT5. We examined the localization of circPRMT5 in MNNG/HOS and MG-63 cells by qRT-PCR and RNA-FISH assay. The results indicated that circPRMT5 was mainly localized in the cytoplasm and not the nucleus (Fig 1H and 1I).

### CircPRMT5 promotes proliferation, migration and invasion and inhibits apoptosis of osteosarcoma cells

To explore the role of circPRMT5 in osteosarcoma cells, we overexpressed and knocked down circPRMT5 in MNNG/HOS and MG-63 cells (Fig 2A). Cell proliferation was evaluated by MTT and colony formation assays. The results showed that overexpression of circPRMT5 increased the proliferative ability of osteosarcoma cells (Fig 2B and 2C). Transwell and wound-healing assays revealed that circPRMT5 overexpression promoted the migratory and

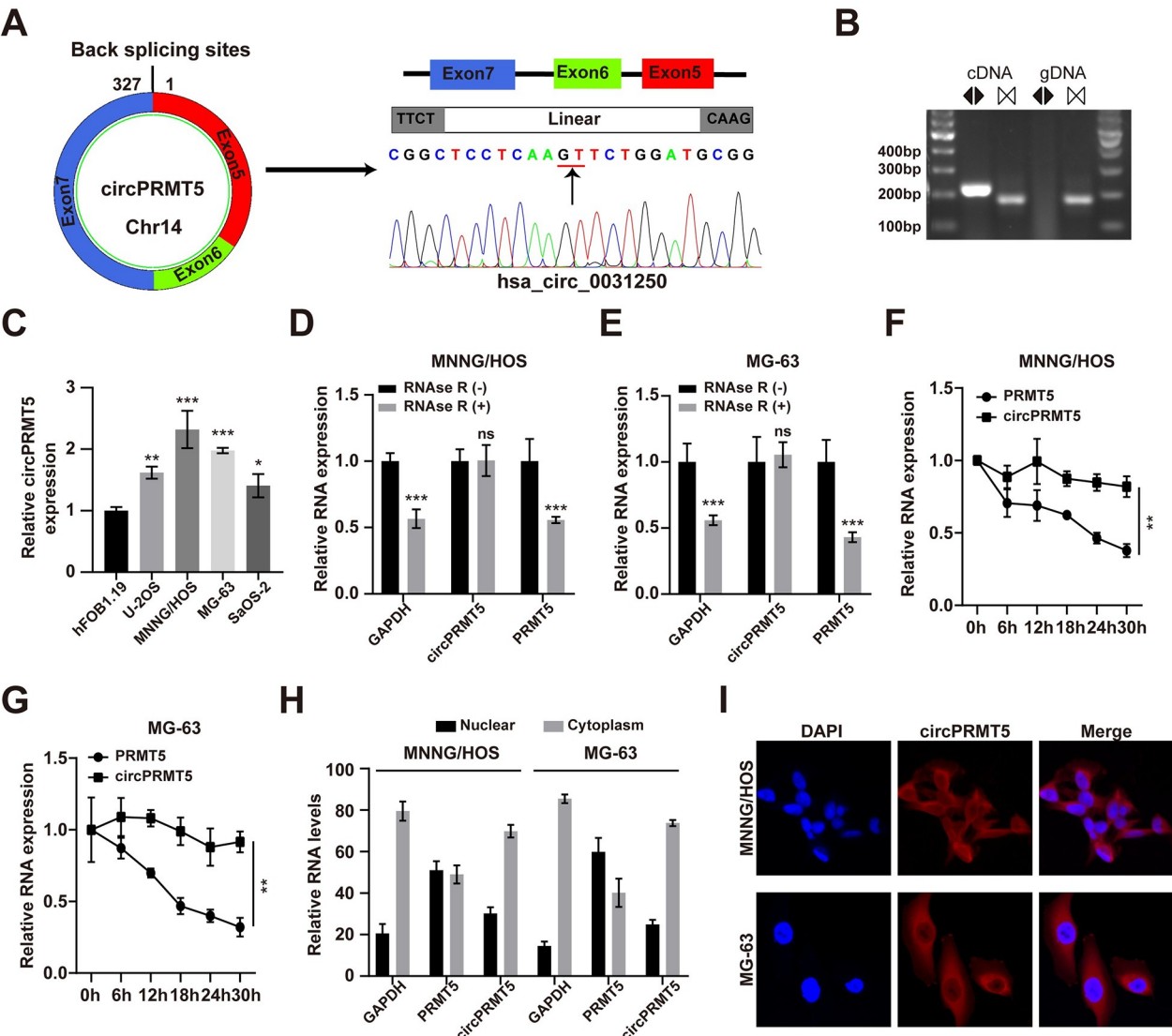

**Fig 1. CircPRMT5 is highly expressed in osteosarcoma cells and mainly located in the cytoplasm.** A. Sanger sequencing results of circPRMT5. B. DNA electrophoresis of the PCR products from divergent or convergent primers. C. qRT-PCR results of circPRMT5 expression in hFOB1.19 cells and osteosarcoma cell lines. D-E. qRT-PCR results of circPRMT5 and PRMT5 expression in MNNG/HOS and MG-63 cells after RNase R treatment. F-G. qRT-PCR analyses of circPRMT5 and PRMT5 expression after actinomycin D (1 μg/ml) treatment for various times. H. qRT-PCR analyses of circPRMT5 and PRMT5 levels in the nuclear and cytoplasmic extracts of osteosarcoma cells. I. RNA-FISH assay with the circPRMT5 probe. Data are expressed as mean ± SD. *p<0.05, **p<0.01, ***p<0.001, ns, not significant.

invasive abilities of osteosarcoma cells (Fig 2D–2F). In contrast, circPRMT5 knockdown significantly inhibited the proliferation, invasion and migration capabilities of osteosarcoma cells (Fig 2B–2F). Moreover, we found that overexpression of circPRMT5 resulted in a decrease in osteosarcoma cells apoptosis and a corresponding increase the percentage of cells in S phase, whereas silencing circPRMT5 had the opposite effects (Fig 3A and 3B). Notably, the expression of cell proliferation markers (MCM2 and PCNA) was significantly increased when circPRMT5 overexpression, and knockdown of circPRMT5 decreased their expression (Fig 3C). Together, our results demonstrated that circPRMT5 might serve as a tumor oncogene to regulate the progression of osteosarcoma cells.

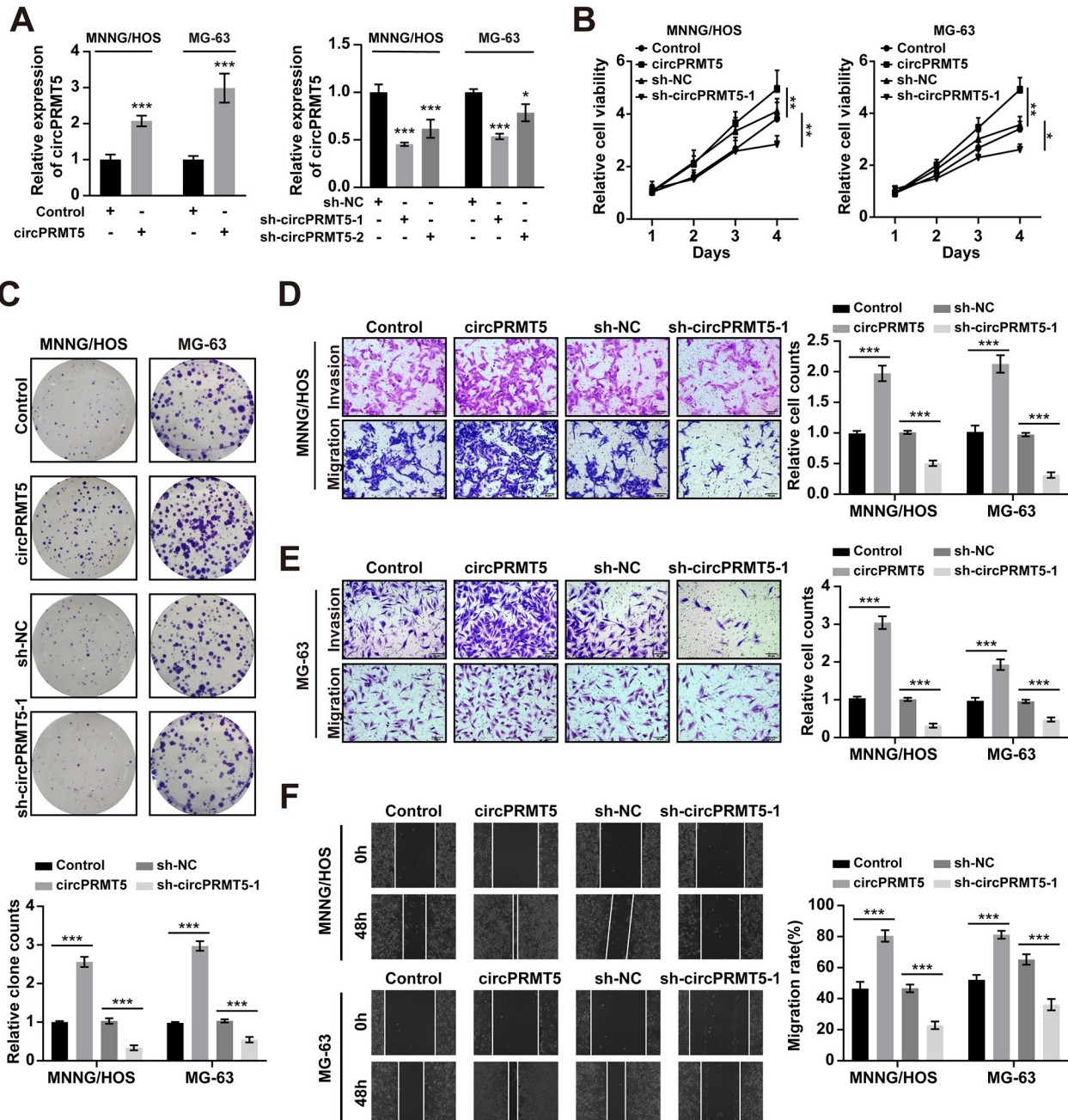

**Fig 2. CircPRMT5 promotes the proliferation, invasion and migration of osteosarcoma cells.** A. The efficacy of circPRMT5 overexpression or sh-circPRMT5 plasmid in MNNG/HOS and MG-63 cells detected by qRT-PCR. B. MTT assay in MNNG/HOS and MG-63 cells after circPRMT5 overexpression or knockdown. C. Colony formation assay in MNNG/HOS and MG-63 cells after circPRMT5 overexpression or knockdown. D-E. Transwell assay in MNNG/HOS and MG-63 cells after circPRMT5 overexpression or knockdown. F. Wound-healing assay in MNNG/HOS and MG-63 cells after circPRMT5 overexpression or knockdown. Data are expressed as mean ± SD. *p<0.05, **p<0.01, ***p<0.001.

## CircPRMT5 binds the RCG domain of CNBP

To further investigate downstream targets of circPRMT5, we first examined the expression of its parental gene and found that enforced circPRMT5 expression did not affect PRMT5 expression level (Fig 4A). RIP assay showed no significant difference in the enrichment of

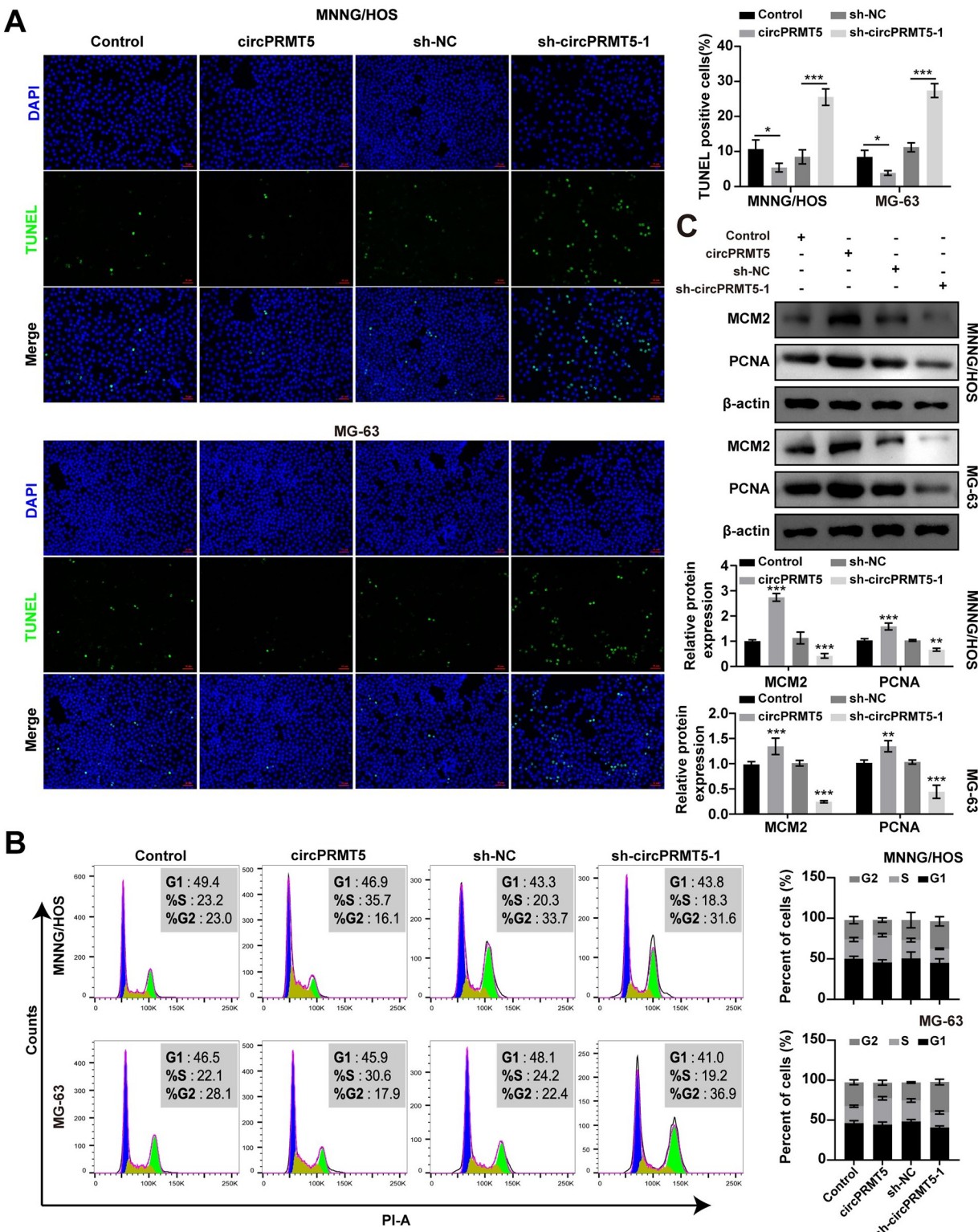

**Fig 3. CircPRMT5 promotes the proliferation and inhibits the apoptosis of osteosarcoma cells.** A. TUNEL assay in MNNG/HOS and MG-63 cells after circPRMT5 overexpression or knockdown. B. The cell cycle distribution of MNNG/HOS and MG-63 cells detected by flow cytometry after circPRMT5 overexpression or knockdown. C. Western blot analysis of MCM2 and PCNA expression in MNNG/HOS and MG-63 cells after circPRMT5 overexpression or knockdown. Data are expressed as mean ± SD. ***p<0.001.

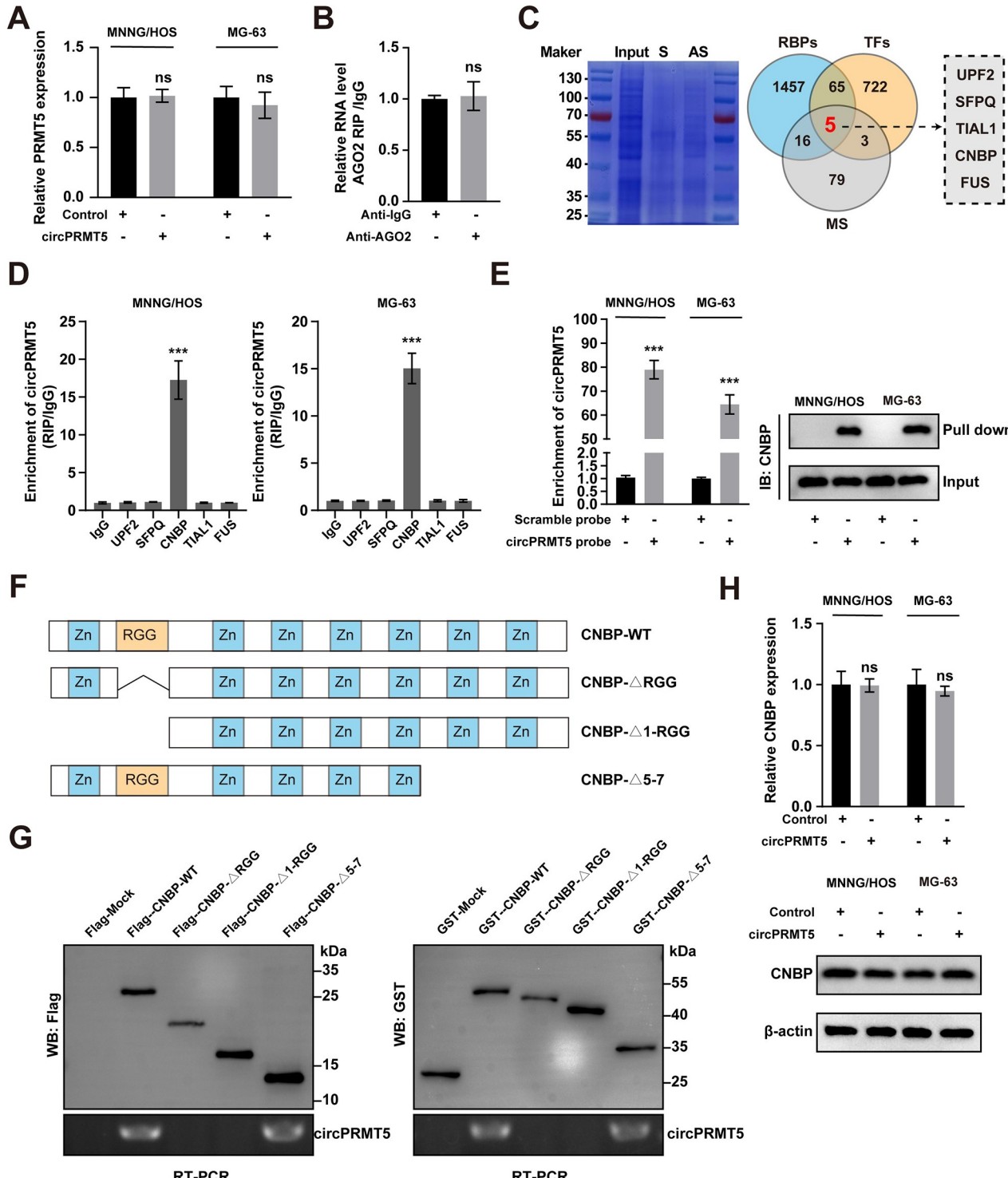

**Fig 4. CircPRMT5 binds with the RCG domain of CNBP.** A: qRT-PCR results of PRMT5 mRNA expression in MNNG/HOS and MG-63 cells after circPRMT5 overexpression. B. qRT-PCR analysis of circPRMT5 enrichment in RIP assay with IgG or AGO2 antibody. C. Overlapping analysis of RBP and TF gene lists with MS results. D. qRT-PCR analysis of circPRMT5 enrichment in RIP assay with IgG or the indicated antibodies. E. RNA pull-down with scramble or circPRMT5 probes in MNNG/HOS and MG-63 cells. F. Schematic of the truncated mutants of CNBP. G. DNA electrophoresis with the PCR products of circPRMT5 immunoprecipitated by the CNBP truncated mutants. H. WB analyses of CNBP expression after circPRMT5 overexpression. Data are expressed as mean ± SD. ***p<0.001, ns, not significant.

circPRMT5 with AGO2 or IgG (Fig 4B), which indicates it may not function as a miRNA sponge.

Previous studies reported that some circRNAs exert their function in cancer development by interacting with RBPs [17]. To explore this possibility, we collected the binding proteins of circPRMT5 using RNA-pulldown assays and identified the interacting proteins with MS. We compared the results with a list of established RBPs and TFs from databases, and five proteins were identified (Fig 4C). RIP assays showed that among the five candidate proteins, only CNBP interacted with circPRMT5 (Fig 4D). RNA-pulldown assay with anti-control or anti-circPRMT5 probe confirmed the interaction between circPRMT5 and CNBP (Fig 4E).

To investigate the specific domain that CNBP binds circPRMT5, we generated three truncated mutants of CNBP (Fig 4F). RIP assays were performed with wild-type or truncated mutants of CNBP. As shown in Fig 4G, the RCG domain of CNBP was required to bind with circPRMT5. We also observed that circPRMT5 overexpression did not affect the expression level of CNBP (Fig 4H).

## CircPRMT5 recruits CNBP to regulate the stability and translation of CDK6 mRNA

To explore the target gene of circPRMT5, we analyzed the differently expressed genes in osteosarcoma from TCGA database and GSE12865. The results were overlapped with the predicted genes of circPRMT5 from the ENCORI database and three genes were selected (Fig 5A and 5B). Among the three genes, only CDK6 mRNA expression was changed after circPRMT5 overexpression or knockdown (Fig 5C). Previous studies reported that circRNAs can promote the binding of RBPs to mRNA, which in turn facilitates the stability and translation of mRNA [26]. Using RIP experiments, we found that overexpression of circPRMT5 promoted the interaction of CNBP and CDK6 mRNA and knockdown of circPRMT5 had the opposite effects (Fig 5D and 5E). Moreover, the half-life of CDK6 mRNA was prolonged in response to overexpression of circPRMT5, and shortened upon knockdown of circPRMT5 (Fig 5F and 5G). We further found that circPRMT5 overexpression resulted in increased translation of CDK6 (Fig 5H and 5I). Overall, these data suggested that circPRMT5 maintained the stability and translation of CDK6 mRNA through promoting the binding between CNBP and CDK6 mRNA.

## CircPRMT5 promotes the malignant activity of osteosarcoma cells by regulating CDK6

To examine whether the function role of circPRMT5 in promoting the malignant activity of osteosarcoma cells involves CDK6, we performed rescue experiments. First, we confirmed the efficacy of CDK6 knockdown by WB (Fig 6A). The results from MTT and colony formation assays supported that CDK6 knockdown blocked the proliferative ability enhanced by circPRMT5 overexpression (Fig 6B and 6C). Moreover, CDK6 knockdown reversed the effects of circPRMT5 overexpression on invasion and migration (Fig 6D–6F). Together, these findings indicated that circPRMT5 promoted osteosarcoma cell malignant activity by upregulating CDK6 expression.

## Discussion

Recently, the role of circRNAs in cancer has been the focus of research attention. Various studies have shown that circRNAs are aberrantly expressed in various cancerous tissues and exert crucial functional role in regulating cancer progression [27]. CircPRMT5 was shown to promote the development of various types of tumors, and it is a promising therapeutic target for

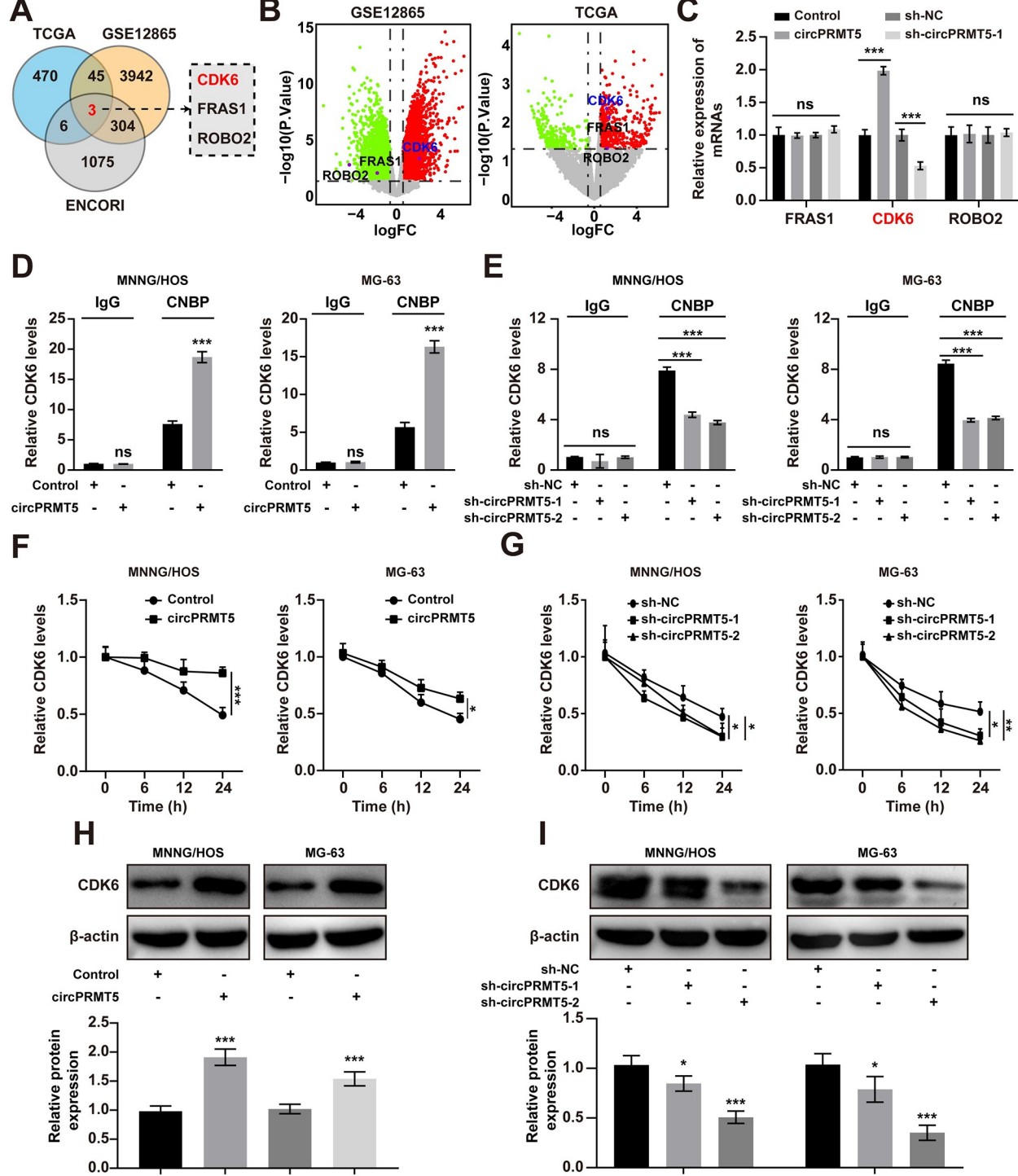

**Fig 5. CircPRMT5 recruits CNBP to regulate the stability and translation of CDK6 mRNA.** A. Analysis of the overlap of the predicted genes from ENCORI database and DEMs from TCGA database and GSE12865. B. Differentially expressed mRNAs in osteosarcoma and normal tissues from GSE12865 and TCGA database. C. qRT-PCR results of the indicated gene expression after circPRMT5 overexpression or knockdown. D. qRT-PCR analysis of CDK6 mRNA immunoprecipitated by CNBP after circPRMT5 overexpression. E. qRT-PCR analysis of CDK6 mRNA immunoprecipitated by CNBP after circPRMT5 knockdown. F. qRT-PCR analysis of CDK6 mRNA expression after actinomycin D (1 μg/ml) treatment for various times after circPRMT5 overexpression. G. qRT-PCR analysis of CDK6 mRNA expression after actinomycin D (1 μg/ml) treatment for various times after circPRMT5 knockdown. H. WB results of CDK6 expression after circPRMT5 overexpression. I. WB results of CDK6 expression after circPRMT5 knockdown. Data are expressed as mean ± SD. $*p<0.05$, $**p<0.01$, $***p<0.001$, ns, not significant.

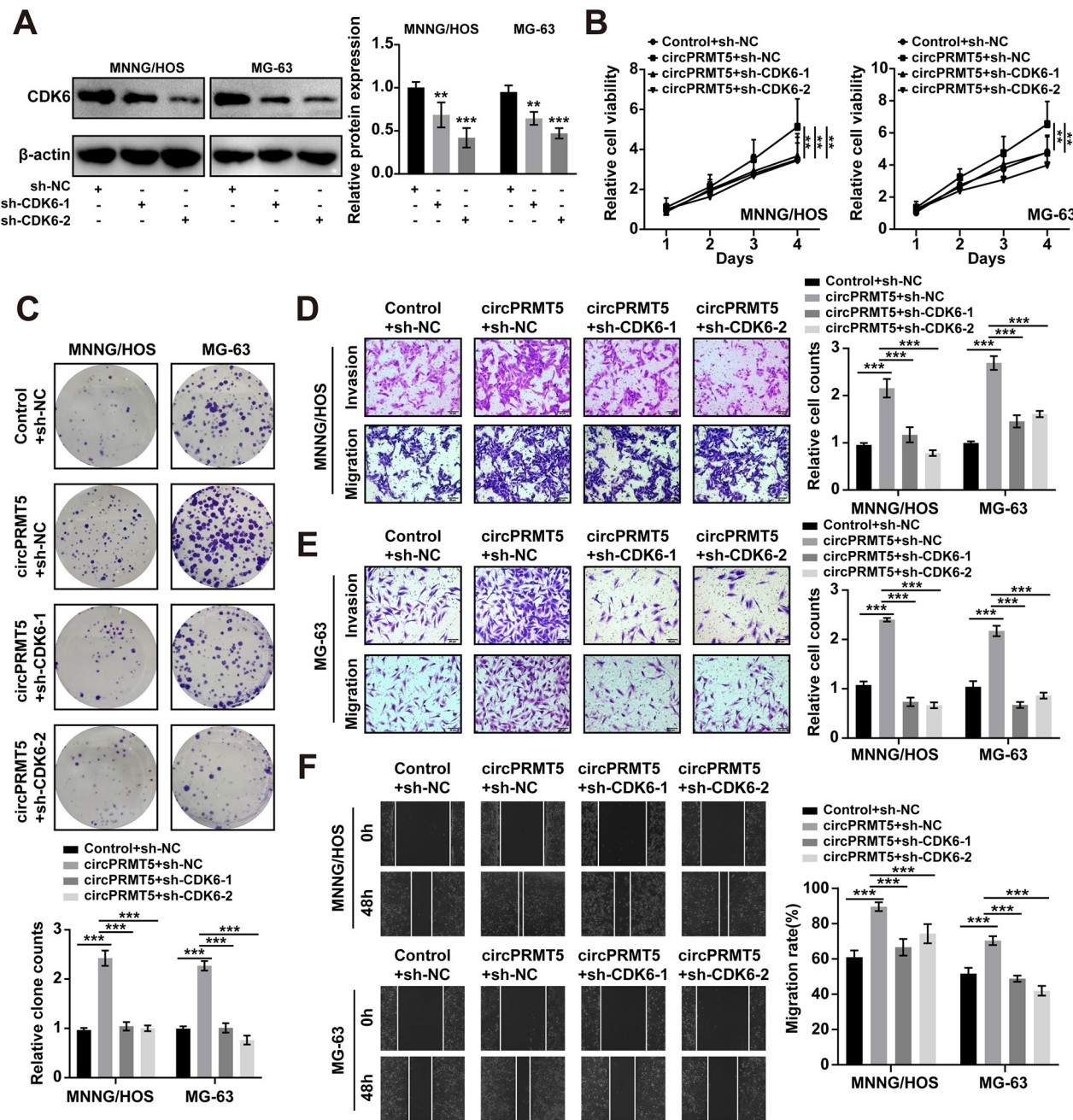

**Fig 6. CircPRMT5 promotes proliferation, invasion and migration of osteosarcoma cells by CDK6.** A. The efficacy of sh-CDK6 plasmid was confirmed by WB. B. MTT assay in MNNG/HOS and MG-63 cells after circPRMT5 overexpression and CDK6 knockdown. C. Colony formation assay in MNNG/HOS and MG-63 cells after circPRMT5 overexpression and CDK6 knockdown. D-E. Transwell assay in MNNG/HOS and MG-63 cells after circPRMT5 overexpression and CDK6 knockdown. F. Wound-healing assay in MNNG/HOS and MG-63 cells after circPRMT5 overexpression and CDK6 knockdown. Data are expressed as mean ± SD. **p<0.01, ***p<0.001.

cancer. However, the role of circPRMT5 in osteosarcoma has been unknown. Here, we revealed that circPRMT5 was significantly upregulated in osteosarcoma and overexpression of circPRMT5 promotes the malignant activity of osteosarcoma cells. Our findings indicate that circPRMT5 may serve as a potential therapeutic target for osteosarcoma.

Many previous studies have focused on the mechanism of circRNAs as a miRNA sponge, in which circRNAs bind to miRNA and release the inhibition on downstream target genes [28]. For instance, Zhang et al. reported that circSTAU2 mitigated its inhibitory effect on CAPZA1 via binding to miR-589 [23]. Recent studies have indicated that circRNAs also act as a crucial regulators of gene expression via interacting with RBPs, which regulate the stability and translation of mRNAs [17]. Herein, we demonstrated a role of circPRMT5 in cancer cells by recruiting an RBP to regulate the translation and stability of a target mRNA. Several studies have indicated that one circRNA can regulate a biological process through different mechanisms [29], indicating that there are still many functions of circPRMT5 in cancer to be explored.

CNBP is a cellular nucleic acid-binding protein. Initial studies on CNBP mainly focused on its role in the embryogenesis of craniofacial structures and the human disease myotonic dystrophy type 2 [30–32]. As an RBP, CNBP is involved in the transcriptional and post-transcriptional regulation of numerous critical genes, and thus participating in many different biological processes [33]. Notably, CNBP is closely associated with tumor occurrence and development [34]. Yang et al. revealed that overexpression of CNBP promoted gastric cancer progression [35]. Consistent with previous studies, our findings suggested that CNBP may act as a mRNA chaperone and enhance the mRNA translation and stability, thus promoting tumor progression.

CDK6 is a key regulator of cell-cycle that plays important role in the occurrence and survival of tumors [36]. Moreover, CDK6 expression is increased in many cancer types, and tumor progression is accompanied by enhanced activity of CDK6 [37,38], suggesting that CDK6 is an attractive target for tumor therapy. CDK4/6 inhibitors have shown good efficacy in clinical practice; these inhibitors have changed the therapeutic strategy for breast cancer and are showing promising effects in other malignancies [39]. Our study suggests that circPRMT5 promotes the malignant activity of osteosarcoma cells by promoting CDK6 expression, and inhibition of CDK6 expression can reverse these stimulative effects. This suggests that CDK6 may be an important downstream effector of circPRMT5 in osteosarcoma.

In conclusion, our results revealed that circPRMT5 promotes osteosarcoma malignant activity by recruiting CNBP to facilitate the translation and stability of CDK6 mRNA. Thus, circPRMT5 may represent a promising therapeutic target for osteosarcoma.

## Supporting information

**S1 Table. Primers sequences for qRT-PCR.**
(DOCX)

**S2 Table. Sequences information for gene knockdown.**
(DOCX)

**S3 Table. Antibody information used in this study.**
(DOCX)

**S1 Raw images.**
(PDF)

**S1 Appendix.**
(DOCX)

**S1 File.**
(DOCX)

## Acknowledgments

We appreciate all participants in this study.

## Author Contributions

**Conceptualization:** Yunlu Liu, Hongyan Jiang, Keli Hu, Weiguo Zhang, Xiaofei Jian.

**Data curation:** Yunlu Liu, Hui Zou, Xiaofei Jian.

**Formal analysis:** Yunlu Liu, Hongyan Jiang, Keli Hu, Hui Zou.

**Investigation:** Keli Hu, Jiangtao Liu, Xiaofei Jian.

**Methodology:** Yunlu Liu, Hui Zou, Xiaofei Jian.

**Project administration:** Jiangtao Liu, Xiaofei Jian.

**Supervision:** Weiguo Zhang, Jiangtao Liu, Xiaofei Jian.

**Validation:** Hongyan Jiang, Keli Hu, Hui Zou.

**Writing – original draft:** Yunlu Liu, Keli Hu.

**Writing – review & editing:** Hongyan Jiang, Weiguo Zhang, Jiangtao Liu, Xiaofei Jian.

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
