## [Decision Letter · Decision Letter 0]

14 Sep 2023

PONE-D-23-20535CircPRMT5 promotes progression of osteosarcoma by recruiting CNBP to regulate the translation and stability of CDK6 mRNAPLOS ONE

Dear Dr. Jian,

Thank you for submitting your manuscript to PLOS ONE. After careful consideration, we feel that it has merit but does not fully meet PLOS ONE’s publication criteria as it currently stands. Therefore, we invite you to submit a revised version of the manuscript that addresses the points raised during the review process.

We look forward to receiving your revised manuscript.

Kind regards,

Zhijie Xu

Academic Editor

PLOS ONE

Journal Requirements:

Reviewers' comments:

Reviewer's Responses to Questions

**Comments to the Author**

1. Is the manuscript technically sound, and do the data support the conclusions?

Reviewer #1: Yes

Reviewer #2: Yes

2. Has the statistical analysis been performed appropriately and rigorously? 

Reviewer #1: No

Reviewer #2: Yes

3. Have the authors made all data underlying the findings in their manuscript fully available?

Reviewer #1: Yes

Reviewer #2: Yes

4. Is the manuscript presented in an intelligible fashion and written in standard English?

Reviewer #1: Yes

Reviewer #2: Yes

5. Review Comments to the Author

Reviewer #1: 1. Please professionally edit the English language, including but not limited to tense and grammar.

2. The declaration “Our study is the first to demonstrate the role of circPRMT5 in tumor by recruiting RBPs to regulate the translation and stability of mRNA.” is arbitrary.

3. Fig. 3A of TUNEL representative figures are too inexplicit to been seen clearly. Please change show the readers high quality pictures.

4. Please provide the statistical analyses for each protein level in Fig. 3C, 5H, 5I, and 6A.

5. Please upload the original result pictures of Fig. 4G.

Reviewer #2: The manuscript tried to explore the role of circPRMT5 in the progress of osteosarcoma. Sanger sequencing results suggested that circPRMT5 was highly expressed in osteosarcoma cells. With multiple bio-techniques, it was found that circPRMT5 promoted the binding of CNBP to CDK6 mRNA. And the authors speculated that circPRMT5 promotes progression of osteosarcoma by recruiting CNBP to regulate the translation and stability of CDK6 mRNA. There is some novelty in this topic, but some points still need to be addressed for the publication in the journal of PlOS ONE.

1. Were there any cell lines acting as controls along with MNNG/HOS and MG-63 cell lines when the knockdown or overexpression were performed?

2. Can some circPRMT5 inhibitors be used to test whether they can inhibit the proliferation of osteosarcoma cells?

3. The relationship between circPRMT5 and CDK6 had better be elaborated. In the context, It was described that knockdown of CDK6 could rescue the proliferative ability enhanced by circPRMT5 overexpression, consequently, it was suggested that CDK6 knockdown could reverse the enhanced effects of circPRMT5 overexpression on invasion and migration, seems a bit contradictory.

6. PLOS authors have the option to publish the peer review history of their article (what does this mean?). If published, this will include your full peer review and any attached files.

Reviewer #1: No

Reviewer #2: No

---

## [Author Response · Author response to Decision Letter 0]

28 Oct 2023

Dear Pro. Zhijie Xu,

R.E.: PONE-D-23-20535

Thank you very much for your supervision of the reviewing process of our manuscript entitled " CircPRMT5 promotes progression of osteosarcoma by recruiting CNBP to regulate the translation and stability of CDK6 mRNA." We also highly appreciate the reviewer’s conscientiousness, carefulness and the broad knowledge on the relevant research field, since they have given us some beneficial suggestions. we have substantially revised our manuscript according to the reviewer’s comments. All amendments are highlighted in red in the revised manuscript. In addition, point-by-point responses to the comments are listed below this letter.

We hope that the revision is acceptable for the publication in your journal.

Look forward to hearing from you soon. 

With best wishes,

Yours sincerely, 

Dr. Xiaofei Jian

First of all, we would like to express our sincere gratitude to the reviewers for their constructive and positive comments.

Replies to Reviewer 1:

Reviewer comment: 1. Please professionally edit the English language, including but not limited to tense and grammar.

Author response: Thanks for your positive comment on the present study and insightful suggestion. We have edited the original manuscript in English language.

Reviewer comment: 2. The declaration “Our study is the first to demonstrate the role of circPRMT5 in tumor by recruiting RBPs to regulate the translation and stability of mRNA.” is arbitrary.

Author response: Thank you for pointing this out. Correction has been made in the revised manuscript.

Reviewer comment: 3. Fig. 3A of TUNEL representative figures are too inexplicit to been seen clearly. Please change show the readers high quality pictures.

Author response: Thank you for pointing this out. Accordingly, the high-quality pictures have been added in Fig. 3A.

Reviewer comment: 4. Please provide the statistical analyses for each protein level in Fig. 3C, 5H, 5I, and 6A.

Author response: We are very sorry for our negligence of this information. The statistical analysis has been performed and the graph has been added in the Fig. 3C, 5H, 5I, and 6A of the revised manuscript.

Reviewer comment: 5. Please upload the original result pictures of Fig. 4G.

Author response: We have therefore uploaded the original uncropped and unadjusted images in the revised manuscript.

Replies to Reviewer 2:

Reviewer comment: 1. Were there any cell lines acting as controls along with MNNG/HOS and MG-63 cell lines when the knockdown or overexpression were performed?

Author response: For investigate the biological roles of circPRMT5 in osteosarcoma cells, we established the overexpression and knockdown systems by using Lv-circRNA or sh-circRNA in MNNG/HOS and MG-63 cell lines. The efficiency of overexpression or knockdown was confirmed by qRT-PCR relative to the negative control. In present study, we did not use other cell lines acting as controls along with MNNG/HOS and MG-63 cell lines when the knockdown or overexpression were performed. In fact, circPRMT5 has been reported to be upregulate in other cancer cell lines (PMID: 30305293; PMID: 32020730), and the overexpression and knockdown systems have also been successfully established in their studies.

Reviewer comment: 2. Can some circPRMT5 inhibitors be used to test whether they can inhibit the proliferation of osteosarcoma cells?

Author response: We appreciate the reviewer to point out this professional point. In the current study of circRNA, construction of shRNA, which covered the back-splicing region of circRNA for silencing, is often used to inhibit its expression in cancer cells. Therefore, to determine whether circPRMT5 could influence the biological functions of osteosarcoma cells, circPRMT5 expression was stable knockdown by transfection with shRNA and sh-NC. The results showed that silence of circPRMT5 could significantly inhibit osteosarcoma cells’ proliferation capacities via MTT and colony formation assays (Fig 2B and C). 

Reviewer comment: 3. The relationship between circPRMT5 and CDK6 had better be elaborated. In the context, It was described that knockdown of CDK6 could rescue the proliferative ability enhanced by circPRMT5 overexpression, consequently, it was suggested that CDK6 knockdown could reverse the enhanced effects of circPRMT5 overexpression on invasion and migration, seems a bit contradictory.

Author response: We felt very sorry that the way we express this sentence is not accurate. Accordingly, we have corrected this sentence in revised manuscript.

---

## [Decision Letter · Decision Letter 1]

18 Dec 2023

PONE-D-23-20535R1CircPRMT5 promotes progression of osteosarcoma by recruiting CNBP to regulate the translation and stability of CDK6 mRNAPLOS ONE

Dear Dr. Jian,

Thank you for submitting your manuscript to PLOS ONE. After careful consideration, we feel that it has merit but does not fully meet PLOS ONE’s publication criteria as it currently stands. Therefore, we invite you to submit a revised version of the manuscript that addresses the points raised during the review process.

We look forward to receiving your revised manuscript.

Kind regards,

Zhijie Xu

Academic Editor

PLOS ONE

Journal Requirements:

Reviewers' comments:

Reviewer's Responses to Questions

**Comments to the Author**

1. If the authors have adequately addressed your comments raised in a previous round of review and you feel that this manuscript is now acceptable for publication, you may indicate that here to bypass the “Comments to the Author” section, enter your conflict of interest statement in the “Confidential to Editor” section, and submit your "Accept" recommendation.

Reviewer #1: All comments have been addressed

Reviewer #2: (No Response)

2. Is the manuscript technically sound, and do the data support the conclusions?

Reviewer #1: Yes

Reviewer #2: Yes

3. Has the statistical analysis been performed appropriately and rigorously? 

Reviewer #1: Yes

Reviewer #2: (No Response)

4. Have the authors made all data underlying the findings in their manuscript fully available?

Reviewer #1: Yes

Reviewer #2: Yes

5. Is the manuscript presented in an intelligible fashion and written in standard English?

Reviewer #1: Yes

Reviewer #2: No

6. Review Comments to the Author

Reviewer #1: The data of this study suggest that circPRMT5 promotes osteosarcoma cell malignant activity by recruiting CNBP to regulate the translation and stability of CDK6 mRNA. Therefore, circPRMT5 may serve as a novel potential therapeutic target for osteosarcoma patients.The authors have completed the revision and the revised manuscript is acceptable now.

Reviewer #2: The manuscript has been revised carefully, there are some minor points need to be addressed.

1. The quality of the figures needs to be improved because they are not very inexplicit.

2. The language had better be improved by some native speakers.

3. Controls had better be performed for the knockdown or overexpression.

7. PLOS authors have the option to publish the peer review history of their article (what does this mean?). If published, this will include your full peer review and any attached files.

Reviewer #1: **Yes: **Xiaojuan Liu

Reviewer #2: No

---

## [Author Response · Author response to Decision Letter 1]

31 Jan 2024

January 30th, 2024

Dear Pro. Zhijie Xu,

R.E.: PONE-D-23-20535R1

Thank you for your letter and the reviewers’ comments concerning our manuscript entitled " CircPRMT5 promotes progression of osteosarcoma by recruiting CNBP to regulate the translation and stability of CDK6 mRNA." Those comments are all valuable and very helpful for revising and improving our paper, as well as the important guiding significance to our researches. We have read through comments carefully and have made corrections. Based on the instructions provided in your letter, we uploaded the file of the revised manuscript. Revised portion are marked in blue in the paper. The responses to the reviewer's comments are presented following.

We would love to thank you for allowing us to resubmit a revised copy of the manuscript and we highly appreciate your time and consideration.

Look forward to hearing from you soon. 

With best wishes,

Yours sincerely, 

Dr. Xiaofei Jian

Reviewer comment: 1. The quality of the figures needs to be improved because they are not very inexplicit.

Author response: Thank you for the suggestion. We have adjusted the quality of the figures to make them clearer and easier to read.

Reviewer comment: 2. The language had better be improved by some native speakers.

Author response: We apologize for the language problems in the original manuscript. The language presentation was improved with assistance from a native English speaker with appropriate research background.

Reviewer comment: 3. Controls had better be performed for the knockdown or overexpression.

Author response: We are grateful for the suggestion. In this study, we established the overexpression and knockdown systems by using Lv-circRNA or sh-circRNA in the osteosarcoma cell lines. The efficiency of overexpression or knockdown was confirmed by qRT-PCR relative to the negative control (Fig. 2A). Moreover, targeting CDK6 with two independent short hairpin RNAs (shRNAs) against the back-spliced junction site resulted in effective knockdown of CDK6. The efficiency of knockdown was confirmed by Western blotting (Fig. 6A). Controls were normally set up in the above experiments. If necessary, we are willing to add additional tests to demonstrate the effect of overexpression or knockdown.

---

## [Editor Report · Decision Letter 2]

2 Feb 2024

CircPRMT5 promotes progression of osteosarcoma by recruiting CNBP to regulate the translation and stability of CDK6 mRNA

PONE-D-23-20535R2

Dear Dr. Jian,

We’re pleased to inform you that your manuscript has been judged scientifically suitable for publication and will be formally accepted for publication once it meets all outstanding technical requirements.

Kind regards,

Zhijie Xu

Academic Editor

PLOS ONE
---

## [Editor Report · Acceptance letter]

4 Apr 2024

PONE-D-23-20535R2 

PLOS ONE

Dear Dr. Jian, 

I'm pleased to inform you that your manuscript has been deemed suitable for publication in PLOS ONE. Congratulations! Your manuscript is now being handed over to our production team.

Kind regards, 

on behalf of

Prof. Zhijie Xu 

Academic Editor

PLOS ONE